# Distinguishing Invasive from Chronic Pulmonary Infections: Host Pentraxin 3 and Fungal Siderophores in Bronchoalveolar Lavage Fluids

**DOI:** 10.3390/jof8111194

**Published:** 2022-11-12

**Authors:** Radim Dobiáš, Pavla Jaworská, Valeria Skopelidou, Jan Strakoš, Denisa Višňovská, Marcela Káňová, Anton Škríba, Pavlína Lysková, Tomáš Bartek, Ivana Janíčková, Radovan Kozel, Lucie Cwiková, Zbyněk Vrba, Milan Navrátil, Jan Martinek, Pavla Coufalová, Eva Krejčí, Vít Ulmann, Milan Raška, David A. Stevens, Vladimír Havlíček

**Affiliations:** 1Department of Bacteriology and Mycology, National Reference Laboratory for Mycological Diagnostics, Public Health Institute in Ostrava, 70200 Ostrava, Czech Republic; 2Institute of Laboratory Medicine, Faculty of Medicine, University of Ostrava, 70300 Ostrava, Czech Republic; 3Department of Biology and Ecology, Faculty of Science, University of Ostrava, 71000 Ostrava, Czech Republic; 4Department of Anesthesiology and Intensive Care Medicine, University Hospital Ostrava, 70800 Ostrava, Czech Republic; 5Institute of Physiology and Pathophysiology, Faculty of Medicine, University of Ostrava, 71000 Ostrava, Czech Republic; 6Department of Intensive Medicine, Emergency Medicine and Forensic Studies, University of Ostrava, 71000 Ostrava, Czech Republic; 7Institute of Microbiology of the Czech Academy of Sciences, 14220 Prague, Czech Republic; 8Department of Medical Microbiology Prague and Kladno, Public Health Institute in Ústí nad Labem, 18600 Prague, Czech Republic; 9Department of Lung Disease and Tuberculosis, University Hospital Ostrava, 70800 Ostrava, Czech Republic; 10Department of Pneumology and Phthisiology, Ostrava City Hospital, 72880 Ostrava, Czech Republic; 11Lung Department, Krnov Combined Medical Facility, 79401 Krnov, Czech Republic; 12Department of Hemato-oncology, University Hospital of Ostrava, 70800 Ostrava, Czech Republic; 13Department of Immunology, Public Health Institute in Ostrava, 70200 Ostrava, Czech Republic; 14Department of Epidemiology and Public Health, Faculty of Medicine, University of Ostrava, 70030 Ostrava, Czech Republic; 15Department of Clinical Biochemistry, AGEL Hornická poliklinika s.r.o., 70200 Ostrava, Czech Republic; 16Department of Microbiology, Faculty of Medicine and Dentistry, Palacky University Olomouc, 77515 Olomouc, Czech Republic; 17Department of Immunology, Faculty of Medicine and Dentistry, Palacky University Olomouc, 77515 Olomouc, Czech Republic; 18California Institute for Medical Research, 2260 Clove Dr., San Jose, CA 95128, USA; 19Division of Infectious Diseases and Geographic Medicine, Stanford University School of Medicine, Stanford, CA 95128, USA; 20Department of Analytical Chemistry, Faculty of Science, Palacký University, 77146 Olomouc, Czech Republic

**Keywords:** invasive fungal disease, pulmonary aspergillosis, bronchoalveolar lavage fluid, pentraxin-3, triacetylfusarinine C, non-neutropenic

## Abstract

The multiple forms of pulmonary aspergillosis caused by *Aspergillus* species are the most common respiratory mycoses. Although invasive, the analysis of diagnostic biomarkers in bronchoalveolar lavage fluid (BALF) is a clinical standard for diagnosing these conditions. The BALF samples from 22 patients with proven or probable aspergillosis were assayed for human pentraxin 3 (Ptx3), fungal ferricrocin (Fc), and triacetylfusarinine C (TafC) in a retrospective study. The infected group included patients with invasive pulmonary aspergillosis (IPA) and chronic aspergillosis (CPA). The BALF data were compared to a control cohort of 67 patients with invasive pulmonary mucormycosis (IPM), non-*Aspergillus* colonization, or bacterial infections. The median Ptx3 concentrations in patients with and without aspergillosis were 4545.5 and 242.0 pg/mL, respectively (95% CI, *p* < 0.05). The optimum Ptx3 cutoff for IPA was 2545 pg/mL, giving a sensitivity, specificity, positive predictive value (PPV), and negative predictive value (NPV) of 100, 98, 95, and 100%, respectively. The median Ptx3 concentration for IPM was high at 4326 pg/mL. Pentraxin 3 assay alone can distinguish IPA from CPA and invasive fungal disease from colonization. Combining Ptx3 and TafC assays enabled the diagnostic discrimination of IPM and IPA, giving a specificity and PPV of 100%.

## 1. Introduction

Invasive fungal diseases (IFD) are one of the important causes of morbidity and mortality in healthcare, especially in patients with multiple comorbidities. This is mainly due to inadequate sensitivity and specificity in clinical tests. Prompt identification and aggressive therapy are essential for successfully treating these opportunistic infections [1].

Invasive pulmonary aspergillosis (IPA), caused by various *Aspergillus* species, is the most common respiratory mycosis in immunocompromised patients, with *Aspergillus fumigatus* the most common causative agent. Invasive pulmonary aspergillosis affects individuals with weakened immunity, especially neutropenic patients [2]. However, the development of IPA in non-neutropenic patients has become more common in recent years [1,3], and differentiating IPA from other lower respiratory tract infections has made the early diagnosis of IPA more challenging [4].

Invasive pulmonary aspergillosis can be diagnosed by targeting either serum or bronchoalveolar lavage fluid (BALF). Several studies have highlighted the value of galactomannan (GM) as a marker of this condition in BALF [5,6], showing that it can achieve a sensitivity of 100% and a specificity of 76%. Moreover, a meta-analysis of published studies on the diagnosis of IPA based on GM in BALF found that proven or probable IPA was distinguished from non-IPA with sensitivities of 84–88% and specificities of 81–88% when applying an optical density index cutoff above 1.0 [6].

Unfortunately, diagnostic tests used routinely to detect biomarkers (GM, 1,3-β-D-glucan, IgA, and IgG specific to *A. fumigatus* and *Aspergillus* DNA) have somewhat limited reliability when applied to samples from a primarily non-sterile site. However, a recent study showed that analyzing fungal siderophores, including ferricrocin (Fc) and triacetylfusarinine C (TafC) in BALF samples, can improve the early diagnosis of IPA because these biomarkers have specificity values of almost 100% [7]. Including siderophores in the diagnosis could improve the diagnostic criteria for BALF samples without requiring a transbronchial biopsy. This conclusion is supported by in vitro and in vivo studies showing that intracellular Fc and extracellular TafC closely correlate with the polarized phase of the fungal growth and indicate the proliferation of the pathogen with angioinvasion [8].

Long pentraxin 3 (Ptx3) is a plasma-soluble receptor that serves as a non-specific human proinflammatory biomarker and is synthesized in the endothelium and macrophages, neutrophils, fibroblasts, and other immune cells [9,10]. Following pathological stimulus during IPA, its concentration in the BALF increases more rapidly than those of short pentraxins (C-reactive protein, serum amyloid P-component) secreted by hepatocytes following IL-6 stimulation. Adequate treatment of the infection reduces the concentration of Ptx3 [9]. There have been two seminal reports describing the use of Ptx3 as a biomarker in the analysis of BALF samples from critically ill patients: Li et al. reported a study involving 35 proven and 16 suspected IPA groups [10], while Kabbani et al. reported fast Ptx3 secretion kinetics in a study involving 15 patients with probable IPA, 30 with lower respiratory tract colonization, and 17 with positive GM and negative BALF culture results [9].

*A. fumigatus* secretes various virulence factors, including siderophores, which mammalian hosts do not synthesize. Moreover, fungal siderophore biosynthesis is a growth phase-dependent process that may reflect true pathogen angioinvasion from lung tissue [8] to the bloodstream and urine [11]. Our research group’s fundamental goal is to develop a combined analysis that simultaneously targets non-specific protein host factors such as Ptx3 and highly specific metallophore fungal biomarkers to improve the early diagnosis of fungal infection. Here we report an essential step towards this goal through the concurrent detection of a pathogen’s siderophores and a host factor.

## 2. Patients and Methods

### 2.1. Study Design and Patient Selection

This retrospective observational study included patients aged >18 years admitted to ICUs and Respiratory Departments in three centers (University Hospital Ostrava, City Hospital Ostrava, and Combined Medical Facility in Krnov) between January 2018 and June 2020 (see the Informed Consent section).

Clinical data and BALF samples were collected from patients initially suspected of having a respiratory tract disease. Their enrollment was based on clinical suspicion of pulmonary aspergillosis (PA) and at least one of the following clinical/laboratory findings: (a) underlying pulmonary conditions such as chronic obstructive pulmonary disease (COPD), emphysema, or bronchiectasis; (b) extrapulmonary conditions such as immunocompromising disorders, diabetes, solid organ malignancies, and corticosteroid therapy; (c) clinical symptoms of lower respiratory tract infection such as fever, productive cough, hemoptysis, or dyspnea that were not alleviated by antibacterial therapy; (d) chest medical imaging (X-ray or CT scan) showing pulmonary infiltrates, lesions, nodules or consolidation changes; or (e) microbiological evidence in the form of Aspergillus spp. BALF culture or BALF galactomannan detection. Another requirement was the availability of BALF samples collected before initiating antifungal therapy.

Twenty-two BALF samples from patients with PA were included in the study. These were divided into three cohorts representing invasive aspergillosis (IPA = 18, 1 proven and 17 of probable IPA), chronic PA (CPA = 4). The non-aspergillosis control cohort consisted of 67 patients, including four with invasive pulmonary mucormycosis (IPM, 4 proven IPM) as well as a group with pulmonary fungal colonization (FCol) and pulmonary bacterial infection (BInf) (Figure 1).

### 2.2. Criteria for Pulmonary Aspergillosis

Diagnosis of IFD was based on the criteria of the European Organization for Research and Treatment of Cancer and the Mycoses Study Group Education and Research Consortium (EORTC/MSGERC) consensus definitions, as well as the guidelines of the European Society for Clinical Microbiology and Infectious Diseases (ESCMID) and European Committee for Medical Mycology for CPA [1,12]. Diagnostic criteria for proven IPA were based on histological, cytopathologic, direct microscopy, or culture evidence of *Aspergil-lus* spp. in specimens obtained by biopsy or sterile procedures using samples from a normally sterile site. In addition, confirmed TafC secretion was used as an indicator of the proliferation of *Aspergillus* hyphae and possible angioinvasion [11].

Probable IPA diagnosis was based on mycological evidence in the form of (a) cytology, direct microscopy, and/or culture evidence indicating the presence of *Aspergillus* spp. in a lower respiratory tract sample; or (b) a GM index above 0.5 in serum and/or above 0.8 in BALF if the following clinical and host factor criteria were satisfied. The clinical criteria were based on the observation of at least one clinical/radiological abnormality consistent with a pulmonary infectious disease process, namely: (a) dense, well-circumscribed lesion(s) with or without a halo sign; (b) air crescent sign; (c) cavitation; (d) wedge-shaped and segmental or lobar consolidation; or (e) detection of tracheobronchial ulceration, pseudomembrane formation, nodules, plaques, or eschar by bronchoscopy. The host factor criteria were: (a) steroid treatment with a prednisone equivalent of 20 mg or more per day; (b) qualitative or quantitative neutrophil abnormality (inherited neutrophil deficiency or an absolute neutrophil count of ≤500 cells/mm^3^); (c) chronic respiratory airway abnormality (COPD, bronchiectasis); (d) decompensated cirrhosis; (e) treatment with recognized immunosuppressants (e.g., calcineurin or mammalian target of rapamycin inhibitors, blockers of tumor necrosis factor and similar antifungal immunity pathways, alemtuzumab, ibrutinib, or nucleoside analogs) during the past 90 days; (f) hematological malignancies/hematopoietic stem cell transplantation; (g) solid organ transplant recipients; (h) human immunodeficiency virus infection; or (i) severe influenza (or other severe viral pneumonia, such as COVID-19).

Diagnosis of CPA was based on (a) chronic respiratory symptoms, including cough, sputum production, hemoptysis, and dyspnea; (b) one or more cavities with or without a fungus ball present or nodules visible upon chest imaging; (c) direct evidence of aspergillosis (microscopy or culture from biopsy) or fungal biomarker evidence, such as positive BALF GM or aspergillus-specific IgG; and (d) histological evidence of *Aspergillus* hyphae in lung biopsy specimens or a positive *Aspergillus* spp. culture from lung biopsy specimens.

Diagnosis of IPM was proven by culture from lung sterile site (tissue, puncture). Culture-positive BALF samples that did not meet the criteria for proven, probable IPA or proven and probable IFD above were considered as colonization (FCol).

### 2.3. Assays for Ptx3, CRP, CREA, IPA, and CPA Diagnosis Biomarker Monitoring

Pentraxin 3 concentrations were measured using an ELISA kit from BioVendor (Czech Republic). Pentraxin 3 concentrations in the BALF were stable during storage at −80 °C for 1 day, 1 week, 3 months, and 6 months. During the study, the mean storage time for a BALF sample was 6 months. C-reactive protein (CRP) was detected using the CardioPhase hsCRP assay and the Atellica^®®^ NEPH 630/BNII/BN ProSpec^®®^ analytical system (Siemens Healthcare, Erlangen, Germany). Galactomannan antigen detection was performed with the Aspergillus EIA kit (Bio-Rad Platelia™) according to the manufacturer’s instructions. The BALF samples were considered positive if the GM positivity index (PI) value was ≥1.0 based on the EORTC/MSGERC definitions [1]. A. fumigatus-specific antibodies were detected using an *A. fumigatus* IgG ELISA kit (Immunolab GmbH, Kassel, Germany). Serum samples were considered positive if the index value of the IgG class was ≥1.2 [3]. Samples were stored at −80 °C and subjected to retrospective Fc and TafC examination by infection metallomics [13]. Because human hosts do not secrete TafC, the cutoff for this marker was defined as the limit of detection (LOD = 0.3 ng/mL) for our high-performance liquid chromatography (HPLC)–Fourier transform ion cyclotron resonance mass spectrometry (FTICR MS) method. We have previously shown this cutoff eliminates false positives [11]. Creatinine (Crea) levels were measured using a creatinine enzymatic assay (Beckman Coulter Ireland Inc., Lismeehan, O’Callaghan’s Mills Co., Clare, Ireland).

### 2.4. Liquid Chromatography and Mass Spectrometry of Siderophores

Triacetylfusarinine C and Fc standards were obtained from EMC Microcollections GmbH (Tübingen, Germany). The BALF samples were subjected to two-step liquid–liquid extraction [14], in which 50 µL samples were spiked with a Ferrioxamine E internal standard (1 µg/mL, 7.5 µL), extracted twice with ethyl acetate, and dried under reduced pressure. The remaining aqueous phase was mixed with four equivalents of methanol and frozen (−80 °C, 1 h). Precipitated proteins were removed by centrifugation (14,000× *g*, 4 °C, 10 min), and the supernatant was transferred to a vial containing the residue from the evaporated ethyl acetate fraction, then concentrated under reduced pressure. The pooled extract was then resuspended in 15% LC-MS-grade acetonitrile and analyzed by HPLC-MS using a Dionex UltiMate 3000 HPLC system (Thermo Fisher Scientific, Waltham, MA, USA) with an Acquity HSS T3 C18 analytical column (1.8 μm, 1.0 × 150 mm, Waters Corporation, Manchester, UK) coupled to a SolariX 12T FTICR mass spectrometer (Bruker Daltonik, Bremen, Germany). One microliter of the extract solution was injected into the HPLC-MS system. Gradient elution of analytes was performed at a flow rate of 50 μL/min using solvents A and B, where B was 95% acetonitrile (ACN), and A was water containing 1% ACN and 0.1% formic acid. The proportion of B was initially held at 2% for 2 min before being raised linearly to 60% over 7 min and then to 99% over 2 min, followed by an isocratic wash with 99% B for 3 min before returning to 2% B over 0.5 min and re-equilibrating with 2% B for 5.5 min. The MS was operated in positive ESI mode. Each sample was analyzed with quadrupole settings of 200–600 (low mass) and 500–1000 (high mass) Da, using appropriate ion transfer optics tuning in both cases. Data were processed qualitatively and quantitatively with our in-house software CycloBranch (version 2) [15] and Bruker Data Analysis 5.0, respectively. The BALF calibration standards were prepared from 50 µL aliquots of pooled control human BALF (10 donors) spiked with commercial TafC (variable concentration, 15 µL). A quantitative analysis of TafC stability and fragmentation behavior was performed as described by Dobias et al. [11].

### 2.5. Statistical Analysis

Analyses of quantitative data were conducted in R 4.0.2 [16] using standard libraries for exploratory data analysis (calculating medians and means and generating boxplots) and statistical testing. The normality of the age and Ptx3 distributions was evaluated using the Shapiro–Wilk test. To compare the age structure between groups, we used the Wilcoxon rank-sum test, assuming equal variance in the proportion of males. Fisher’s exact test was used to compare sample frequencies. Due to the data distribution of Ptx3, the significance of differences in Ptx3 levels between groups was evaluated using the nonparametric Kruskal–Wallis test and the post hoc pairwise Wilcoxon signed-rank test. A statistical significance threshold of *p* < 0.05 was applied in all cases. Cutpointr, an R package for tidy calculation of “optimal” cutpoints” [17], was used to draw receiver operating characteristic (ROC) curves and analyze specificity and sensitivity. The ROC curve (receiver operating characteristic curve) is a probability curve; it is a graph showing the performance of a classification model at all classification thresholds. This curve plots two parameters: True Positive Rate and False Positive Rate. The AUC (area under the ROC curve) represents the degree or measure of separability and measures the entire two-dimensional area underneath the entire ROC curve from (0.0) to (1.1) [17,18]. We used several methods to estimate cutoff values (maximize spline, gam, locally estimated scatterplot smoothing, LOESS), estimates for metric accuracy and the sum of sensitivity and specificity, and kernel estimation for the Youden’s index. False negative and false positive rates were calculated as controls based on probable PA and non-PA.

## 3. Results

### 3.1. Chronic Obstructive Pulmonary Disease and Steroid Treatment Are Common Risk Factors for Pulmonary Aspergillosis

Of the 89 patients enrolled into this retrospective study, 22 had some form of PA, while the remaining 67 served as controls (Table 1). Chronic obstructive pulmonary disease and corticosteroid therapy represented the most serious risk factors in the PA cohort, occurring in 46 and 50% of PA patients, respectively. Both risk factors exhibited high statistical significance—*p* < 0.001 for COPD and *p* < 0.038 for corticosteroid therapy.

### 3.2. Pentraxin 3 in BALF Exhibits High Specificity for Invasive Aspergillosis and Mucormycosis

As shown in Figure 2, the median Ptx3 concentrations determined in BALF samples from PA and non-PA patients differed significantly (4545.5 pg/mL and 242.0 pg/mL, respectively; *p* < 0.05). In addition, the median Ptx3 concentration in the IPA cohort (4613.5 pg/mL) was more significant than that in the CPA cohort (1746.0 pg/mL; *p* = 0.05). The non-PA subgroups (IPM, Fcol, and BInf) also clustered distinctly (*p* < 0.05) and had median Ptx3 concentrations of 4326.0, 242.0, and 224.5 pg/mL, respectively. Unfortunately, it was not possible to distinguish IPA from IPM based on Ptx3 levels. However, the Ptx3 assay effectively distinguished invasive fungal infection (IPA/IPM) from fungal colonization and IPA from CPA.

The BALF Ptx3 threshold for the combined detection of IPA or IPM was 1653 pg/mL, with a sensitivity and specificity of 96.2% (95% CI = 80–100%) and 87.3% (95% CI = 77–94%), respectively, according to the ROC curve analysis (AUC = 0.9795, see source calculation in Appendix A). According to the ROC analysis, the optimal BALF Ptx3 cutoff for IPA was 2545 pg/mL. When applying this cutoff, the sensitivity and specificity were 100% (95%CI 81–100%) and 98% (95%CI 91–100%), respectively (AUC = 0.9991, Appendix A). A Ptx3 concentration above the diagnostic cutoff value was detected in one CPA sample, and a GM index value above the cutoff was also detected in one CPA sample. On the other hand, all four patients with proven IPM had high Ptx3 concentrations together with sub-threshold GM index values. This observation could potentially enable the early discrimination of invasive aspergillosis from mucormycosis, although more detailed studies with larger patient cohorts would be needed to verify this possibility. Notably, CRP, as another host factor, was not detected in any IPA or non-IPA BALF samples (Table 2).

### 3.3. Fungal Siderophores in BALF

Table 2 shows the frequencies at which Fc and TafC concentrations above the corresponding cutoff values were detected in different patient groups and the corresponding values for Ptx3 and GM. Pentraxin 3 remains the most successful biomarker with 100% sensitivity for IPA and IPM, while GM exhibited a fair sensitivity of 78%. Triacetylfusarinine C (normalized to creatinine) also showed a reasonable sensitivity of 61%; in cases where creatinine data were missing due to lack of concurrent specimen (*n* = 15), the mean creatinine over the patient’s dataset concentration was taken instead. Intracellular Fc, which is an indicator of probable fungal cell lysis [20], was present in seven BALF samples and exhibited 28% sensitivity for IPA.

The negative control patient group included patients with bacterial infections (one *Acinetobacter* sp., two *Citrobacter* sp., three *Escherichia* sp., four *Haemophilus* spp., five *Klebsiella* spp., one *Morganella* sp., six *Mycobacterium* spp., two *Staphylococcus* sp., one *Streptococcus* sp.). Neither Fc nor TafC was detected in any bacterial or fungal colonization (nineteen *Candida* spp., six *Aspergillus* spp., eight *Penicillium* spp., two *Schizophyllum* spp., one *Cladosporium* sp., one *Trichoderma* sp.) controls other than in one suffering from co-infection of *Acinetobacter baumanii, A. fumigatus,* and *Pseudomonas aeruginosa*. The combined use of TafC and Ptx3 as biomarkers of *A. fumigatus* infection offered no improvement in critical diagnostic parameters when compared to Ptx3 alone (Table 3). However, TafC and Fc were not present in any IPM patients and thus warranted further clinical study as potentially discriminatory biomarkers (Appendix A). In addition, TafC had a specificity and positive predictive value of 100% when distinguishing IPA from IPM (Appendix A).

## 4. Discussion

The optimum Ptx3 cutoff for IPA was 2545 pg/mL, giving a high sensitivity, specificity, PPV, and NPV (Table 3). Similar to the IPA group, the median Ptx3 concentration for IPM was 4326 pg/mL. Pentraxin 3 assay can distinguish IPA from CPA and invasive fungal disease from colonization. In addition, combining Ptx3 and TafC assays enabled the diagnostic discrimination of IPM and IPA, giving a specificity and PPV of 100%.

The Ptx3 levels in BALF may consistently be elevated in patients with a chronic underlying disease and critically ill patients. In such patients, Ptx3 concentrations above 2545 pg/mL may indicate the onset of IPA or IPM. The AUC values for these IPDs were 0.9795 and 0.9991, respectively (Appendix A), which exceeded the value of 0.91 reported for BALF samples by Li et al. [10]. The value of Ptx3 as an early biomarker is supported by Kabbani et al. (2017), who found that Ptx3 levels rose much more quickly than CRP levels in response to inflammation or infection.

In agreement with the literature [3,10], the most common underlying condition in our *Aspergillus-*infected patient population was COPD (overall incidence: 45.5%), supporting the conclusion that COPD is a risk factor for pulmonary aspergillosis. *A. fumigatus* can be isolated from up to 20% of COPD patients [2]. It should also be noted that Pauwels identified cigarette smoke as a trigger of Ptx3 biosynthesis in the pulmonary veins of mice [21], and similar Ptx3 production mechanisms may exist in humans. Moreover, Ptx3 is a non-specific acute phase reactant produced by various cells in response to local tissue injury or inflammation [10,22]. Consequently, elevated Ptx3 levels may not always reflect the severity of the infectious component in IPA patients. Chronic obstructive pulmonary disease patients may also have elevated levels of Ptx3 in both plasma and BALF compared to healthy subjects. Chronic obstructive pulmonary disease is a chronic airflow restriction that worsens over time and is linked to an increased chronic inflammatory response with severe extrapulmonary consequences upon exposure to various noxious stimuli [22,23]. It was, therefore, essential to verify that the elevated levels of Ptx3 in BALF due to COPD or other factors unrelated to IPA could not lead to false positive IPA diagnoses. Our results show that COPD patients have higher Ptx3 levels than control cohorts but not to the degree that could cause the misdiagnosis of fungal infections, if our proposed cutoffs are utilized. More patients with COPD in the control group would have made such a conclusion more definitive. However, Ptx3 concentration does not correlate with COPD severity [22]. Thus, it is highly probable that the Ptx3 results in COPD patients were not affected.

Interestingly, up to 30% of IPA patients are treated with corticosteroids, a known risk factor for IPA [1,2]. This raises the question of whether corticosteroids affect Ptx3 production. However, this possibility was rejected by consecutive screening studies on 41 patients showing that prednisolone did not impact Ptx3 levels in plasma [24]. In future prospective IFD cohort studies, it would be important to show that BALF Ptx3 concentrations are also not influenced by corticosteroid or other immunosuppressive treatments.

The detection of *Aspergillus* siderophores [11,19] and elevated Ptx3 can discriminate aspergillosis from mucormycosis. Using TafC as a representative *Aspergillus* siderophore, we confirmed the 61% presence of PA in all PA-positive samples (Appendix A). This is notable given that most of the samples examined in this work were analyzed retrospectively, up to 3 years after collection. One false positive detection of *A. fumigatus* was found in the control group of 67 patients. The false positivity in this specific patient we may explain by gastro-esophageal reflux, which is a known cross-contamination pathway [25].

Although our study was retrospective, we retrieved data showing that all 18 patients with probable IPA and 4 with CPA in our cohort over time showed a favorable response to voriconazole therapy, by clinical examination, laboratory assessment and/or imaging. Of note, in the IPA group, negative correlations were observed between GM and Ptx3 (Appendix A) and between GM and both siderophores (Appendix A). Prospective clinical trials are needed to confirm the results presented herein and to evaluate the possibility of discriminating between aspergillosis and mucormycosis by simultaneously analyzing biomarkers of host metabolism (Ptx3) and pathogen activity (siderophores), especially due to the small number of IPM patients in our study.

## 5. Conclusions

Invasive fungal infections, including IPA, can be diagnosed in just a few hours by monitoring Ptx3 levels in BALF [26]. If Ptx3 assay is combined with siderophore detection in BALF, IPA can be confirmed with very high probability (PPV = 94% and NPV = 95%), making it possible to distinguish IPA from IPM (PPV = 100% and NPV = 100%). When supported by a positive BALF GM assay, invasive aspergillosis can be diagnosed with very high confidence, 95% and 92% of Ptx3 and TafC, respectively.

## Figures and Tables

**Figure 1 jof-08-01194-f001:**
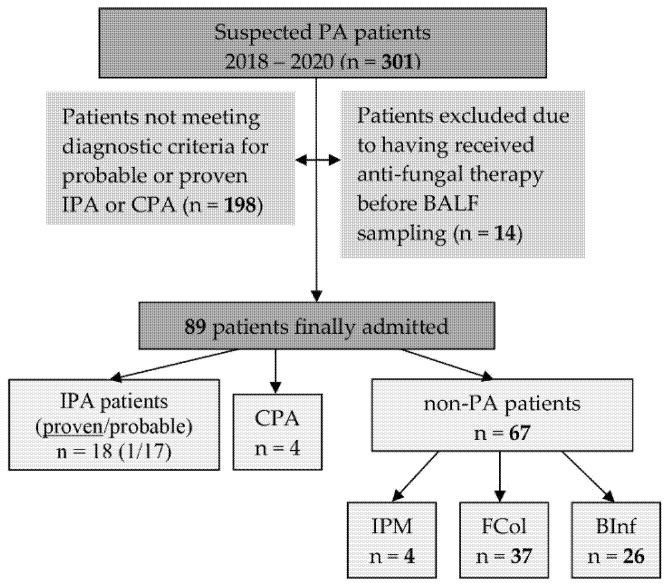
Flow chart showing the process used to select patients for inclusion in the analysis. PA: Pulmonary Aspergillosis; IPA: Invasive Pulmonary Aspergillosis; CPA: Chronic Pulmonary Aspergillosis; IPM: Invasive Pulmonary Mucormycosis; FCol: pulmonary Fungal Colonization; BInf: pulmonary Bacterial Infections.

**Figure 2 jof-08-01194-f002:**
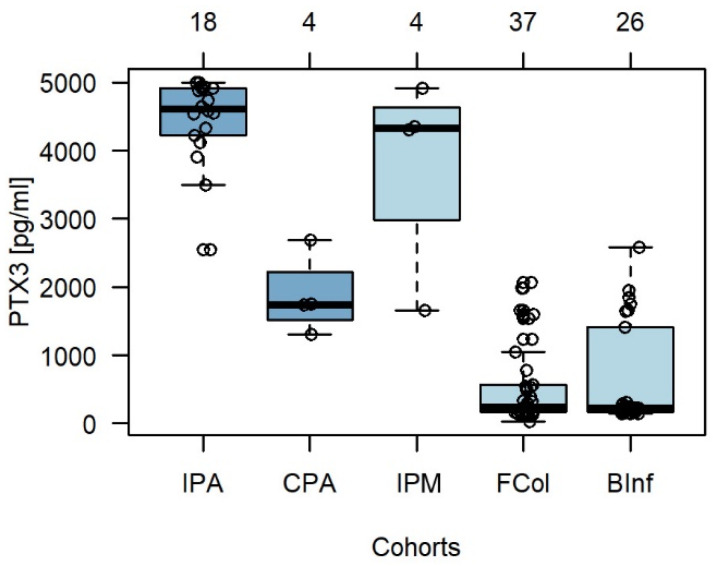
Bronchoalveolar lavage fluid (BALF) pentraxin 3 (Ptx3) concentrations in patients with aspergillosis and control cohorts (based on Appendix A). Ptx3, pentraxin 3; IPA, Invasive Pulmonary Aspergillosis; CPA, Chronic Pulmonary Aspergillosis; IPM, Invasive Pulmonary Mucormycosis; Fcol, Fungal Colonization; BInf, pulmonary Bacterial Infections.

**Table 1 jof-08-01194-t001:** Characteristics of patients (*n* = 89) with pulmonary fungal disease and the control group.

	PA (*n* = 22)	Non-PA (*n* = 67)	*p*
Age (mean, range)	59 (45–79)	61 (24–85)	0.361
Male (ratio)	13 (59%)	43 (68%)	0.800
ICU	14 (64%)	11 (16%)	**<0.001**
**Underlying pulmonary diseases:**			
COPD	10 (46%)	2 (3%)	**<0.001**
Bronchopneumonia	3 (14%)	22 (33%)	0.104
IPF	0 (0%)	10 (15%)	0.062
H1N1 influenza	2 (9%)	0 (0%)	0.059
Asthma	0 (0%)	11 (16%)	0.059
Bronchitis	0 (0%)	5 (8%)	0.327
Bronchiectasis	0 (0%)	3 (5%)	0.572
Mycobacteriosis	0 (0%)	1 (2%)	1.000
Sarcoidosis	0 (0%)	3 (5%)	0.572
COVID-19	1 (5%)	1 (2%)	0.436
**Extrapulmonary diseases:**			
SOM	1 (5%)	3 (5%)	1.000
Haemato-oncology	2 (9%)	2 (3%)	0.254
Cardiovascular disease	0 (0%)	1 (1.5%)	1.000
AKI	1 (5%)	0 (0%)	0.247
TPL	1 (5%)	2 (3%)	1.000
Polytrauma	0 (0%)	1 (2%)	1.000
Osteoarthritis	1 (5%)	0 (0%)	0.247
Steroid treatment	11 (50%)	16 (24%)	**0.038**

PA: Pulmonary aspergillosis; ICU: Intensive Care Unit; COPD: Chronic Obstructive Pulmonary Disease; IPF: Interstitial Pulmonary Fibrosis; SOM: Solid Organ Malignancy; AKI: Acute Kidney Injury; TPL: Organ transplantation.

**Table 2 jof-08-01194-t002:** Concentrations of TafC and Fc in BALF from patients with aspergillosis and control cohorts.

	Fc/Crea (ratio *, >0)	TafC/Crea (ratio *, >0)	Ptx3 (≥2545 pg/mL)	CRP (>0.17 mg/L)	GM (PI ≥ 1)
**Pulmonary Aspergillosis**					
IPA (*n* = 18)	5 (28%)	11 (61%)	18 (100%)	0 (0%)	14 (78%)
CPA (*n* = 4)	1 (25%)	0 (0%)	1 (25%)	0 (0%)	2 (50%)
**Invasive Pulmonary Mucormycosis**					
*Rhizopus* spp. (*n* = 4)	0 (0%)	0 (0%)	4 (100%)	0 (0%)	0 (0%)

IPA: Invasive Pulmonary Aspergillosis; CPA: Chronic Pulmonary Aspergillosis; Fc: ferricrocin; TafC: triacetylfusarinin C; Ptx3: pentraxin 3; CRP and GM: C-reactive protein and galactomannan in bronchoalveolar lavage fluid; Crea: creatinine; * “creatinine index” [19]; PI: Positivity Index.

**Table 3 jof-08-01194-t003:** Biomarker cutoffs with the highest diagnostic significance for Invasive Pulmonary Aspergillosis.

	Ptx3 (≥2545 pg/mL) (CI %)	TafC/Crea (ratio ^†^) (CI %)	Fc/Crea (ratio ^†^) (CI %)	Ptx3-TafC * (CI %)	GM (PI ≥ 1) (CI %)
Sensitivity (%)	100 (81–100)	61 (36–83)	0 (0–6)	81	71 (42–92)
Specificity (%)	98 (91–100)	98 (91–100)	100 (81–100)	98	97 (89–100)
PPV (%)	95 (74–100)	92 (62–100)	0 (0–100)	94	83 (52–98)
NPV (%)	100 (94–100)	90 (80–96)	22 (14–33)	95	94 (85–98)
AUC	0.9991	0.8007	0.3668	0.8181	0.7051

* Ptx3-TafC—if both tests are positive, i.e., Ptx3 ≥ 2545 pg/mL and TafC ≥ 0.3 ng/mL. Fc: ferricrocin; TafC: triacetylfusarinine C; Ptx3: pentraxin 3; Crea: creatinine; † “creatinine index” [19]; GM: galactomannan; AUC: Area Under the Curve (Receiver operating characteristic analysis).

## Data Availability

Not applicable.

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
