# Peer review of "Distinguishing Invasive from Chronic Pulmonary Infections: Host Pentraxin 3 and Fungal Siderophores in Bronchoalveolar Lavage Fluids"

_jof, 2022, doi:10.3390/jof8111194_

Round 1

Reviewer 1 Report

According to the study was done retrospectively, the sera were collected 3 years before the study was carried out, therefore, I consider that there is a high probability that the Ptx3 levels will decrease, which could in some way lead to false negative results or that there is some other type of interferent that could give false positives. I would rethink the study and take into account the pros and cons of having done it retrospectively. As a test study I consider that the idea is not bad, but since more trials are needed and do it ambispectively comparing those 3 years ago and 3 years in the future and see if there is or not variation in the results.

Author Response

We thank this reviewer for rising this important point.  Please find the attached file.

Reviewer 2 Report

This manuscript details the retrospective analysis of 3 bronchoalvolar lavage tests to improve the discrimination of invasive pulmonary aspergillosis from other infections.

I think this is an important manuscript that merits publication with a few minor changes/clarifications:

Results:

Line 134, meant to read two cohorts?

Given the low numbers in some cohorts, consider plotting all individual results (and medians) in figure 2.  Additionally figure 2 is not referenced in the text.

It would be useful to know how IPM and FCol groups were confirmed as such?

I would like to see full data re: sensitivity and specificity and confidence intervals for Fc and TafC.

I suggest putting panels onto supplementary tables, eg S2 into panel A-C,or separating into S2.1 etc. and adjust figure legends appropriately.

Statistical analysis:

I am unclear why the  Wilcoxon signed ranks test were used, as I believe there were no paired samples?

Discussion:

The study is retrospective, and presumably the samples were stored in -80, is this equivalent to clinical practice?  More discussion of limitations (including number of patients) needs to be made. Particularly conclusions around discrimination between IPA and IPM given the very small numbers of IPM.

Author Response

We are grateful for the overall positive tone of the review report. Below, please find our point-by-point responses to your comments. The manuscript modifications have been performed in MS Word track changes mode. Thank you for critical reading, comments, and recommendations. Thanks to the criticisms, we could improve our manuscript both in clarity and fluency. Please, find the attached file.

With kind regards,

Radim Dobiáš, on behalf of all contributors

Reviewer 3 Report

The manuscript reports on the utility of Ptx3 and fungal siderophore biomarkers in the diagnosis of invasive aspergillosis, using a retrospective design where the levels of these markers are tested on previously frozen patient BAL samples. This is an interesting paper and follows on from previous work by the authors' group and others in assessing the use of these markers in fungal diagnostics, an important area not least due to the current lack of accurate and rapid fungal diagnostic tests. Overall, the manuscript is well written.

I have a few specific comments:

Line 66: I’m not sure it is reasonable to say that IFD are one of the leading causes of mortality and morbidity in healthcare. While they are clearly an important and often neglected cause of disease, I wouldn’t go as far as saying they are a “leading cause” in terms of number of cases.

Line 134: should this not be 2 cohorts (instead of 3 as is written)?

Line 136: It would be useful if the authors could elaborate on their criteria for categorizing patients as having fungal colonization as opposed to disease. 

Line 138: Section 2.2 describes the criteria used for diagnosing patients with aspergillosis, namely the EORTC for invasive aspergillosis and mucormycosis, and the ESCMID criteria for CPA. Much of this section is a description of these criteria, which probably isn't necessary as all the relevant info is contained in the first sentence of the section, so I think this section can be considerably shortened. 

Figure 1: If available, it would be interesting to present the data from samples where patients received antifungals prior to BAL, as often antifungals are commenced on suspicion of fungal infection prior to BAL in real world situations, although I appreciate these data may not be available.

It appears the assays were performed on frozen BAL samples. Do the authors have any data on the effect of freeze/thaw cycle on the results obtained for the Ptx3 and fungal siderophore biomarkers? 

For patients with invasive fungal disease, what proportion had proven and probable disease? It would be useful to report this. 

What proportion of patients in each group were ICU patients? I think it would be important to report this in table 1 as this could be a potential confounder of the results of these assays, particularly Ptx3.

Line 333: "Our results show that COPD patients have higher Ptx3 levels than control cohorts but not to the degree that could cause misdiagnosis of fungal infections, if our proposed cutoffs are utilized." It is not clear to me how the authors have demonstrated this. There appear to be only 2 patients in the control group who had COPD. I would have thought there would need to be a larger number of COPD patients in the control group to enable the authors to confidently state this. 

While the authors have rightly considered the potential for steroid use to be a confounder, and reference a paper showing that steroid use does not seem to increase plasma ptx3 levels, it would be important going forward to show that BAL ptx3 levels are also not influenced by steroid use 

Author Response

(The authors gave the same response as above.)

Reviewer 4 Report

Dear Authors, please:

1- in Table 2: remove the values with 0 and add a sentence describing that to the results section, (Control groups I and II)

2-Please make table 3 shorter and add AURC curve, if possible, you can write the CI in the parenthesis,

3- Please save the first paragraph of the discussion to summarize and discuss your own main results and use them for further discussion

4- Please move the second paragraph from the conclusion to the discussion 

5- Please avoid giving references to tables in the conclusion and mainly use your paper’s conclusion 

Author Response

(The authors gave the same response as above.)

Round 2

Reviewer 1 Report

According to what was sent by the authors, I consider that they justify very well what they were asked for, so the article can be published as it is at present.